# Could Oxidative Stress Play a Role in the Development and Clinical Management of Differentiated Thyroid Cancer?

**DOI:** 10.3390/cancers15123182

**Published:** 2023-06-14

**Authors:** Maria Kościuszko, Angelika Buczyńska, Adam Jacek Krętowski, Anna Popławska-Kita

**Affiliations:** 1Department of Endocrinology, Diabetology and Internal Medicine, Medical University of Bialystok, 15-274 Bialystok, Poland; 2Clinical Research Center, Medical University of Bialystok, 15-274 Bialystok, Poland

**Keywords:** thyroid cancer, oxidative stress, antioxidant capacity, cancer progression

## Abstract

**Simple Summary:**

Increased oxidative stress (OS) has been implicated as a relevant risk factor for cancer development and progression. The incidence of differentiated thyroid cancer (DTC) has risen in several populations globally over the last few decades, accounting for approximately 1% of all cancer cases. Patients with DTC diagnosis have been characterized by an increased OS status, which is associated with a poorer prognosis. Therefore, assessing OS status could potentially be considered a useful tool in DTC clinical management. This measurement could be particularly valuable in personalizing treatment protocols and determining new potential medical targets. The purpose of this article is to evaluate research that examines the influence of OS on the progression of DTC and investigate the feasibility of utilizing OS measurements in DTC risk assessment. Gaining a comprehensive understanding of the correlation between carcinogenesis and OS, specifically in the DTC cases, can facilitate the development of novel therapeutic strategies.

**Abstract:**

Increased oxidative stress (OS) has been implicated as a relevant risk factor for cancer progression. Furthermore, patients diagnosed with differentiated thyroid cancer (DTC) have been characterized by an increased OS status. Therefore, assessing OS status could potentially be considered a useful tool in DTC clinical management. This measurement could be particularly valuable in personalizing treatment protocols and determining new potential medical targets to improve commonly used therapies. A literature review was conducted to gather new information on DTC clinical management, with a particular focus on evaluating the clinical utility of OS. These meta-analyses concentrate on novel approaches that employ the measurement of oxidative-antioxidant status, which could represent the most promising area for implementing clinical management.

## 1. Introduction

Thyroid cancer (TC) makes up roughly 1% of all cancer cases. TC can be classified into three primary histological types: differentiated TC (DTC), which emerges from follicular cells, comprising papillary (PTC) and follicular (FTC) types; undifferentiated TC, which includes poorly differentiated (PDTC) and anaplastic TC (ATC); and medullary TC (MTC), which arises from parafollicular C cells [1]. Over 90% of thyroid malignancies originate from follicular cells. The incidence of DTC has risen in several populations globally over the last few decades, possibly due to the more frequent utilization of ultrasound examinations and fine-needle aspiration biopsies (FNAB) [1,2].

In 2020, approximately 586,000 individuals around the globe received a diagnosis of TC, and 43,800 new cases were reported in the United States in 2022 [3]. Although DTC can occur at any age, the typical age of diagnosis is approximately 50 years old. Additionally it is seven times more prevalent in females than males [3,4,5]. There was a notable rise in mortality based on incidence in the year 2021 due to TC, with an overall rate of 1.1% per year. Moreover, the mortality rate based on incidence for advanced-stage PTC is 2.9% annually [6,7,8].

In the past few years, there has been significant progress in comprehending the genetic and metabolic mechanisms responsible for the risk factors and management of TC, such as radioactive iodine therapy (I-131, RAI). Knowledge of the specific genetic mutations involved is crucial for the management and prognosis of TC. While guidelines for therapeutic management, particularly I-131 treatment, are updated annually, they still contain several ambiguities and require standardization. Therefore, it is necessary to examine the disruptions in metabolic pathways that play a role in the development of DTC.

The aim of this article is to assess research that investigates the impact of oxidative stress (OS) on the progression of DTC and investigate the feasibility of utilizing OS measurement in DTC risk assessment.

## 2. The Diagnosis of Thyroid Cancer

The most critical tool in diagnosing this malignancy is FNAB. Multiple staging systems are used for the risk classification of DTC, based on parameters such as the size of the primary tumor, histological examination, tumor-cell infiltration outside the thyroid capsule, and age at diagnosis [9]. In September 2022, the 5th edition of the World Health Organization (WHO) Classification of Thyroid Neoplasms was revised. The current classification categorizes thyroid tumors according to the size of the primary tumor, histological examination, infiltration of tumor cells beyond the thyroid capsule, and age at the time of diagnosis [9]. The 2022 WHO categorization is noteworthy for introducing a grading system that operates on two levels, which distinguishes high-grade cancers from well-differentiated follicular cell-derived carcinomas and MTCs.

Histologic criteria cannot always differentiate benign from malignant lesions. In this categorization, neoplasms differentiated from follicular cells are classified into three sections: benign tumors, low-risk neoplasms, and malignant neoplasms. Initially, the benign category included follicular nodular disease (FND), follicular adenoma, follicular adenoma with papillary architecture, and oncocytic adenomas of the thyroid in the updated classification. Tumors that are classified as extremely low-risk with a favorable prognosis include non-invasive follicular thyroid neoplasms with papillary-like nuclear features (NIFTP), thyroid tumors of uncertain malignant potential (TT-UMP), and hyalinizing trabecular tumors (HTT) [9].

The revised classification now includes a new tumor type called “high-grade follicular cell-derived carcinomas”, which has two histologic subtypes: traditional poorly differentiated thyroid carcinoma (PDTC) and a new subtype known as “differentiated high-grade thyroid carcinoma (DHGTC)”. DHGTC can originate from PTC, FTC, or oncocytic carcinoma (OCA). Furthermore, the invasive encapsulated follicular variant of papillary thyroid carcinoma (IEFVPTC) is currently recognized as a separate entity rather than a subtype of PTC. IEFVPTC has a mutational and transcriptomic profile similar to that of follicular adenoma (FA) and FTC, which is characterized by RAS mutations, whereas classic PTC and the infiltrative follicular subtype of PTC are associated with BRAF**^V600E^** mutations. Although encapsulated, IEFVPTC has the potential to invade blood vessels within the capsule, leading to distant metastasis [10,11]. In the classification of TC, aggressiveness should be taken into consideration, which is determined by various factors including the Ki-67 index, based on the mitotic count and presence of necrosis. The Ki-67 index is a measure of the percentage of cells within a tumor that are actively dividing, based on the expression of the Ki-67 protein. It is calculated by counting the number of cells that stain positively for Ki-67 and expressing it as a percentage of the total number of cells counted [12]. The Ki-67 index is often used as a prognostic and predictive marker in TC as well. It is useful in identifying high-grade FTC and MTC, and is an important prognostic parameter for predicting the clinical course of the disease [9,13,14,15,16].

It is important to note that even though certain PTCs may display aggressive characteristics, they could still have a less severe clinical progression. Usually, these tumors appear in young individuals and have encapsulated or well-defined characteristics, along with a lower Ki-67 score [17,18].

## 3. The Main Genes Implicated in the Development of Thyroid Cancer

The occurrence of TC is linked to genetic and environmental predispositions, as well as the effects of these factors on oxidative DNA damage [19]. Genetic mutations are present in over 70% of DTC cases [20]. The primary and most common genes that are involved in the pathogenesis of TC are the B-type Raf kinase gene (BRAF), Rat Sarcoma Gene (RAS gene), rearranged during transfection gene (RET gene), the neurotrophic receptor-tyrosine kinase 1 gene (NTRK1), the paired box gene 8-peroxisome proliferator activated receptor gamma gene (PAX8-PPARγ), the p53 gene, the beta-catenin (β-catenin) encoding gene, the adenomatous polyposis coli gene (APC), the growth factor receptor involved in myogenesis-1 gene (GRIM-1), the phosphatase and tensin homolog gene (PTEN), the mesenchymal epithelial transition factor gene (MET), the C-erbβ-2 gene, and epigenetic alterations (Figure 1).

The BRAF gene is a member of the Raf kinase family, which participates in the transduction of growth signals through protein kinases. The BRAF gene produces a protein from the serine/threonine kinase family, which plays a role in transmitting signals along the signaling pathway of mitogen-activated protein kinases (MAPK)/Erk pathway, which affects cellular replication, differentiation, secretion, and apoptosis (Figure 2) [21,22]. Thyroid follicular cells demonstrate significant expression of the BRAF protein, alongside hematopoietic cells, neurons, and testicles [23]. Interestingly, more than 30 BRAF gene mutations have been found to be associated with human cancers. The substitution of thymine with adenine at the nucleotide results in the replacement of valine with glutamate at codon V600E in the activation segment, a mutation that is observed in human cancers and accounts for 90% of cases [24]. As mentioned above, the BRAF gene mutation most often affects codon 600 (BRAF**^V600E^**) and has been identified in approximately 40–70% of PTC [25]. This genetic variation is specific to PTC and has not been detected in any other type of DTC or MTC [26]. BRAF mutations are commonly associated with RET/PTC rearrangement and other frequently occurring genetic mutations, and are thought to play an independent role in the development of cancer. Rusinek et al. showed that the BRAF mutation initiated the development of PTC in transgenic mouse studies [27]. Zhu et al. indicated that detecting the BRAF**^V600E^** gene during the FNAB is more sensitive and accurate in diagnosing TC [28]. Xing et al. conducted a collaborative study which showed a significant association between the BRAF mutation and an increased risk of lymph node metastasis, invasion beyond the thyroid capsule, advanced disease progression and disease relapse [26]. In contrast to the findings of Xing et al., Scheffel et al. contended that the prognostic significance of the BRAF**^V600E^** mutation in PTC is disputed, and has limited influence on the evaluation of risk and the prognosis of PTC patients [29].

The RAS family proteins, encoded by the NRAS, KRAS, and HRAS genes, act as binding enzymes and hydrolyze GTP-GTPases, which are involved in signal transduction and the activation of signaling pathways such as MAPK and PI3K/AKT/mTOR. These proteins regulate the expression of genes related to proliferation, growth, adhesion, cell migration, and apoptosis, and play an important role in thyroid tumorigenesis (Figure 2) [30]. The frequency of mutations that lead to RAS activation is roughly 20 to 25% across all human tumors, with some types of cancer, notably pancreatic cancer, having rates as high as 90% [31]. The frequency of RAS mutations in TC varies according to the origin of tumor cells, ranging from 0% to 57% [32]. Activating mutations within these genes, specifically in exons 2, 3, and 4, are directly related to the development of cancers of the colon, pancreas, lungs, and thyroid gland. Due to its potent oxidative properties, hydrogen peroxidase (H_2_O_2_) is responsible for the high levels of oxidative DNA damage observed in cancerous thyroid tissue and the hyperactivation of MAPK and PI3K/Akt, which mediate Erk signaling [33]. In the case of TC, genetic mutations are mainly located in codon 61 of the 3rd exon of the NRAS gene (occurring in about 20–30% of cases), while mutations in the KRAS and HRAS genes occur in approximately 10% of all cases [25].

The RET protein belongs to the class of transmembrane tyrosine kinase receptors, involved in transmitting signals along the MAPK and PI3K-PTEN-AKT pathways of cellular signaling [34]. Somatic alterations in the RET gene have been identified in conditions such as Hirschsprung’s disease, PTC, multiple endocrine neoplasia (MEN) 2A and MEN B, as well as familial MTC [35]. The RET gene consists of 21 exons, which primarily express two isoforms of 1072 (short isoform) and 1114 (long isoform) amino acids by alternative splicing in the 3′ region. It encodes a receptor tyrosine kinase for members of the glial cell line-derived neurotrophic factor (GDNF) family of extracellular signaling molecules [36,37]. The GDNF family receptor α1 (GFRα1), which is a glycosylphosphatidylinositol-anchored cell-surface protein, is required for GDNF binding. The GDNF-GFRα1 complex mediates RET dimerization [38]. In the presence of the GDNF-GFRα1 dimeric complex, two molecules of RET are recruited to the lipid raft, which brings the cysteine domains of RET close together. This close proximity promotes the formation of a dimer and activates tyrosine kinase signaling [39]. Due to the fact that the RET gene is mainly expressed in C cells producing calcitonin, somatic mutations mainly affect MTC. So far, researchers have identified 15 distinct types of RET/PTC rearrangement, with RET/PTC1, RET/PTC2, and RET/PTC3 being the most commonly occurring [40]. There is a strong correlation between the location of the point mutation and the clinical presentation of the disease. The management of the disease and patient survival depend on the site of the genomic mutation.

PAX8-PPARγ is a gene composed of 11 exons, responsible for encoding the cyclic adenosine monophosphate (cAMP)-dependent protein kinase type 1-alpha regulatory subunit. This kinase is the main mediator of cAMP actions for various cellular processes, including cell differentiation, proliferation, and apoptosis [41]. The PAX8-PPARγ gene codes for each R subunit (PRKR1A, PRKR1B, PRKR2A, PRKR2B) and each C subunit (PRKACA, PRKACB, PRKACG, PRKX), respectively [42]. Mutations in PAX8-PPARγ were linked to a more severe form of TC and a poorer prognosis [43].

Over 50% of human cancers exhibit functional mutations in the p53 gene, as indicated by genetic studies [44,45,46,47]. The p53 gene, which is situated on the short arm of chromosome 17, is classified as a gene that suppresses tumor growth. The protein that this gene encodes is composed of multiple functional domains, such as an amino-terminal transactivation domain, a central DNA binding domain, a carboxy-terminal oligomerization domain, and a proline-rich sequence for recognition [48]. A common polymorphism in humans involves the substitution of an arginine for a proline at codon position 72 of exon 4. The p53 gene is involved in DNA repair activation, initiating apoptosis, and is necessary for the senescence response to short telomeres. In the case of DTC, p53 mutations are located in the region between exons 5 and 8, and have been found in up to 40% of PTC and 22% of FTC cases [49,50]. Therefore, p53-targeted therapy currently represents one of the most promising approaches in the development of anticancer drugs [51].

The role of the MET receptor tyrosine kinase in tumor proliferation is well documented [52,53]. The MET oncogene encodes a non-traditional receptor tyrosine kinase with diverse functions, which can promote neoplastic transformation by enabling cancer cells to persist through physical interactions with adaptor proteins and cooperation with surface receptors that are structurally related [54]. In response to growth factors, cancer cells can proliferate by a regulated process that necessitates specific transcription factors, including the MET ligand, and is managed by various extracellular signals [55]. MET dysregulation is a shared characteristic of various human malignancies, including but not limited to kidney, stomach, thyroid, and brain cancers.

Thyroid cells are generally considered to have a relatively high tolerance to OS compared to many other cell types. However, it is important to note that this does not mean they are completely resistant to OS. The thyroid gland contains several enzymatic antioxidants that help neutralize ROS. Genetic alterations in antioxidant enzymes can impact an individual’s vulnerability to TC. For instance, specific gene polymorphisms responsible for encoding antioxidant enzymes such as glutathione peroxidase (GPx), glutathione reductase (GRd), and superoxide dismutase (SOD) have been linked to a heightened risk of developing TC [56]. These genetic variations can influence the enzymes’ activity or expression, potentially compromising the effectiveness of the antioxidant defense system.

## 4. Treatment of Differentiated Thyroid Cancer

The primary treatment for DTC, regardless of the histopathological type of the tumor, is total thyroidectomy with lymphadenectomy of the cervical region. The use of RAI treatment in DTC should be determined based on the patient’s cancer risk stratification (Table 1). While the advantages of RAI in treating high-risk DTC patients are widely acknowledged, there is still debate about its effectiveness for intermediate and low-risk patients. In 2022, the European Thyroid Association (ETA) released new guidelines for the indications for post-surgical RAI therapy in DTC [57]. The administration of RAI, in accordance with the guidelines of the ETA, American Thyroid Association (ATA), Society of Nuclear Medicine and Molecular Imaging (SNMMI), and European Association of Nuclear Medicine (EANM), is performed for remnant ablation, adjuvant treatment, or treatment of advanced DTC [57,58]. Adjuvant therapy with RAI is used to eliminate potential microscopic foci of TC after complete surgical resection of locoregional, distant, or both of these types of metastases. The goal of this treatment is to minimize the risk of tumor recurrence and progression, and improve survival [59,60,61]. RAI therapy is used as adjuvant treatment or for treating known disease in patients who are classified as high risk for recurrence or who have known structural disease. In such cases, high activities of RAI (≥3700 MBq-100 mCi) are preferred over low activities [57]. For patients in the intermediate-risk category, RAI therapy may be necessary and should be customized based on individual cases. Additionally, RAI ablative treatment results in the destruction of postoperative remnants, enabling easier monitoring of Tg concentration in follow-up observation. The actual dose used will be determined by the treating physician based on appropriate postoperative histopathological evaluation using TNM classification, serum thyroglobulin (Tg) concentration, neck ultrasound of thyroid remnant volume, and the results of postoperative neck and whole-body scintigraphy with I-131 [59]. For ablation of the thyroid remnant after surgery, a dose of 30–100 mCi (millicurie) is typically used. For the treatment of metastatic TC, higher doses ranging from 100–200 mCi or more may be required. Patients who are elderly or who have aggressive histologies, extracapsular extension of the tumor, or multiple N1 and/or lymph node metastases can derive the greatest benefit from post-operative RAI therapy. For patients with minimal extrathyroidal invasion, microscopic or few lymph node metastases, and intrathyroidal PTC with BRAF^**V600E**^ mutation, the decision on whether to undergo RAI therapy can be based on post-operative Tg concentration and neck ultrasound [57,62]. In patients categorized as low-risk, the usefulness of RAI therapy is a topic of ongoing debate, and the decision regarding the use of this treatment should be made considering the patient’s specific risk factors [57]. While some authors suggest that even patients with non-metastatic microcarcinomas may benefit from RAI therapy, others find no significant benefit [63,64].

Current research and recommendations indicate that activities of 1110 MBq-30 mCi are as effective as higher activities for ablating presumably benign thyroid remnants and could be used. In a 10-year follow-up of a randomized, prospective study comparing the recurrence rate of low-risk patients treated with either 1110 MBq-110 mCi or 3700 MBq-100 mCi, both groups showed a similar rate of persistent disease [65].

## 5. Thyroid Cancer and Oxidative Stress

Oxidative stress is a biochemical state characterized by an excessive presence of reactive oxygen species (ROS) and reactive metabolites, which can negatively affect many biological components. ROS are created during typical metabolic activities in cells and are indispensable for inducing signaling pathways that respond to modifications in intra- and extracellular environmental factors [66,67]. These processes can increase membrane permeability and intracellular calcium cation concentration, stimulate the expression of nitric oxide synthase (NOS), and activate the phospholipase A2 enzyme (PLA-2). PLA-2 catalyzes the release of arachidonic acid, which, along with enzymes such as cyclooxygenase and lipoxygenase, is involved in the production of hydroxyl and superoxide radicals, contributing to OS [68,69,70]. Oxidative modification of lipids, proteins, and DNA leads to disturbances in homeostasis and cell death as a result of apoptosis or necrosis [71]. Throughout evolution, the cells of living organisms have developed mechanisms to prevent or reduce the negative effects of free radicals. Intracellular antioxidant systems include both micromolecular antioxidants such as vitamins C and E, coenzyme Q, carotenes, and glutathione, as well as macromolecular antioxidants such as catalase (CAT), GPx GRd, and SOD [72].

Cellular stress can take the form of endoplasmic reticulum stress, mitochondrial stress, OS or a combination of all three. In vivo studies have shown that thyroid cells are more susceptible to cell damage resulting from OS than other organs [73]. The normal function of thyroid cells, including cell proliferation and hormone production, depends on the balance between ROS production and antioxidant systems; an imbalance may contribute to TC initiation and progression [74,75,76,77,78]. Several studies indicate that increased OS levels have been detected in patients, even before cancer was diagnosed [79,80]. In their study, Wang et al. found a compelling association between OS and TC, indicating that high levels of OS may increase the risk of developing, and the progression of, thyroid tumors [80]. OS has also been demonstrated to be a significant independent positive predictor of tumor invasion and lymph-node infiltration, which may be a reason for the occurrence of metastasis [81]. Furthermore, the total oxidative status (TOS) level was lower in patients with TC than in controls [80,82]. There are many biomarkers, such as malondialdehyde (MDA), 8-hydroxy-2-deoxyguanosine (8-OHdG), and antioxidative enzymes (CAT, GPx, GRd, SOD), that are used to evaluate the involvement of OS in cancer pathophysiology [83,84,85].

Szanto et al. reviewed that patients with TC exhibit higher levels of basal genome damage associated with oxidative stress when compared to controls [86]. In a separate study, Maier et al. investigated the mRNA expression of two DNA repair genes, Apurinic/Apyrimidinic Endodeoxyribonuclease 1 (APEX1) and 8-oxoguanine DNA glycosylase (OGG1), in mouse cells derived from the thyroid, liver, lung, and spleen. They observed that APEX1 is more highly expressed in thyroid and liver tissues compared to the lung and spleen, whereas OGG1 expression is slightly higher in the thyroid and lung [73]. Upregulated OGG1 nuclear expression nuclear levels of OGG1 were found in benign and malignant lesions compared to normal thyroid tissue [87]. The findings indicate that the persistent exposure to chronic OS during TC development can have harmful consequences. According to Muzza et al., observed statistically significant differences in the levels of OS in FTC and PDTC/ATC compared to in PTC cases, indicating that there may be a connection between ROS levels and tumor aggressiveness [88]. The study of Martinez-Cadenas et al. suggested that OS caused by inflammation may be responsible for mutations of the BRAF gene, consequently leading to the development of the carcinogenic process [89]. In PTC tumors, the BRAF**^V600E^** mutation has been shown to have a positive correlation with the upregulation and activation of the mTOR pathway [90]. Chronic exposure to OS in TC is predicted to contribute to DNA damage, including base lesions and single-strand breaks, which may result in driver mutations in BRAF and RAS [91]. It has been discovered that BRAF and RAS oncogenes can upregulate OS status. On the other hand, ROS are involved in DNA damage and aging induced by BRAF**^V600E^** [92].

There was a direct relationship consistently observed in DTCs between ROS generation and the aggressiveness of the tumor, as indicated by ATA risk and stage, suggesting that OS may be associated with a more severe tumor presentation [88]. The role of OS in TC development has been characterized by examining the expression of GPx1 and thioredoxin reductase (TrxR1) in TC cells. GPx stands for glutathione peroxidase, which is an enzyme that plays a vital role in the body’s system for protecting against oxidative damage. GPx aids in safeguarding cells against oxidative damage by facilitating the conversion of H_2_O_2_ and lipid peroxides into less-harmful substances. Thyroid H_2_O_2_ is generated by a member of the family of NADPH oxidase enzymes (NOX-es), termed dual oxidase 2 (DUOX2) (Figure 3).

These byproducts are produced during routine cellular metabolism and can potentially harm cell membranes, proteins, and DNA. Maintaining cellular redox balance is a critical function of GPx, as it helps reduce ROS levels and counteracts OS [93].

Metere et al. found a decreased expression of GPx1 and TrxR1 proteins in TC cells compared to healthy cells, which was associated with increased levels of ROS in tumor tissue [94]. Akinci et al. reported a decrease in glutathione levels, a cofactor for GPx, in patients with TC [95]. Several studies have shown significantly lower levels of GPx, GST, SOD, and CAT in TC patients. In their study, Sekhar and al. demonstrated that the use of GPx4 inhibitors resulted in ferroptosis and elevated levels of ROS, halted the migration of tumor cells, increased DNA damage, and inhibited both the mTOR pathway and the DNA-damage repair mechanism in in vitro experiments with PTC cells [96]. The study also revealed that nearly half of the human DTC specimens showed elevated expression of GPx4 compared to control tissue. Increased expression of GPx4 is linked to poorer overall survival [97].

8-hydroxy-2-deoxyguanosine is a biomarker of OS and DNA damage. It is a modified nucleotide that is formed when ROS attack DNA, causing oxidation of the guanine base. It is also considered to be one of the most reliable and widely used biomarkers of oxidative DNA damage. When 8-OHdG is formed, it can cause mutations in the DNA sequence during DNA replication. These mutations can potentially lead to cancer and other diseases. Therefore, the measurement of 8-OHdG levels is important for evaluating the extent of oxidative DNA damage and the risk of developing diseases associated with it [98]. Increased levels of nuclear 8-OHdG were detected in both benign and malignant thyroid tissue, as compared to normal thyroid tissue [87]. Tabur et al. reported significantly higher levels of 8-OHdG in pre-operative DTC patients, and after surgery, a decreasing trend was observed [99]. These observations indicate a relationship between the detrimental effects of long-term exposure to chronic OS and the development of TC.

Protein thiol groups and ferric reducing antioxidant power are important components of the antioxidant defense system in the body. Protein thiol groups are thiol (−SH)-containing groups that are present in various proteins and play a role in maintaining their structure and function. Ferric reducing antioxidant power (FRAP) is a measure of the ability of a substance to reduce ferric ions to ferrous ions and is commonly used as an indicator of antioxidant capacity. The concentration of protein thiol groups and ferric reducing antioxidant power showed a decrease, while there was a drastic increase in the levels of protein carbonyls and myeloperoxidase activity (MPO) observed after thyroidectomy in TC patients [100].

These results suggest an imbalance between free radicals and antioxidants, with an excess of radicals leading to OS and consequently to TC development. These disturbances were still observed 20 days after thyroidectomy [100].

MDA, or malondialdehyde, is a reactive aldehyde that arises from the oxidation of lipids. This process involves the attack of polyunsaturated fatty acids (PUFAs) in cell membranes by ROS, such as free radicals. This results in the formation of lipid radicals that react with molecular oxygen to form peroxides, which then break down to form MDA and other aldehydes. MDA is a highly reactive molecule that can react with proteins and DNA, leading to cellular damage and dysfunction. Research has indicated that levels of MDA are increased in patients with TC, particularly in individuals with advanced or metastatic disease [75,101]. Therefore, MDA levels may serve as a potential biomarker for disease advancement and outcome prediction in these patients. However, some authors have not found a correlation between MDA and the development of TC [102,103].

## 6. RAI Therapy and Oxidative Stress

The radioactive iodine used in RAI therapy is a type of iodine that emits radiation, which selectively targets and destroys thyroid tissue. RAI therapy is frequently employed as a treatment after thyroidectomy for DTC, and has been shown to improve outcomes and increase the likelihood of long-term survival. Studies have reported high 10-year survival rates of up to 90% in DTC patients treated with RAI therapy after surgery, although the actual survival rates can vary depending on various factors such as the stage and type of the cancer, patient age, and other health conditions [104]. Tumor cells are more vulnerable to ionizing radiation than normal cells, making the radiation effect useful in the therapeutic process of DTC [105,106]. However, the use of I-131 in these patients can result in an escalation of ROS production, which can induce OS and disturb the redox balance. The generation of OS by ionizing radiation can lead to irreversible damage to the DNA in the cell’s nucleus [107,108,109]. Exposure to ionizing radiation can cause different forms of molecular harm to DNA, including but not limited to single-strand and double-strand breaks, damage to individual bases, and cross-linking between DNA and proteins [110]. Certain types of harm can be mended by the cellular repair mechanisms. However, since these mechanisms are inefficient, mutations or cell death may occur, making OS a major inducing factor [108,111].

The levels of lipid peroxidation products and antioxidant molecules in DTC patients have been studied both before and after RAI therapy. However, the findings are not unequivocal as different parameters have been used to assess the intensity of OS. Additionally, the time elapsed between the application of RAI and measurement of the intensity of OS and antioxidant status varies from several days to several months. Vrndic et al. conducted a study on patients with DTC who received RAI therapy (3.7 GBq-100 mCi in 11 patients and 5.5 GBq-150 mCi in 9 patients) and showed that three days after RAI therapy, the concentrations of MDA were found to be higher than the pretreatment values [112]. This suggests that RAI therapy can induce OS and lipid peroxidation in patients with DTC, which can lead to cellular damage and contribute to the side effects associated with the therapy. However, the study also found that the total antioxidant status (TAS) values in the DTC group did not change significantly after I-131 therapy. These observations suggest that the relatively stable TAS values in these patients indicate good protection against OS induced by high activity of I-131. This may be due to the activation of the antioxidant defense system in response to the OS caused by I-131 therapy. Nevertheless, more research is needed to fully understand the mechanisms involved in the antioxidant response to RAI therapy and to optimize the therapy for DTC patients.

Buczyńska et al. demonstrated significant changes in OS biomarkers before, 5 days after, and one year after RAI for DTC patients. They observed high serum levels of MDA directly after I-131 treatment and a reduction in this parameter after one year of observation. The authors concluded that the determination of OS biomarkers could allow for the personalization of therapy, including an individually selected activity of RAI and the possibility of reducing side effects [113].

This suggests that the measurement of OS biomarkers can be useful in optimizing RAI therapy for DTC patients, potentially reducing the risk of adverse effects and improving treatment outcomes. Leoni et al. demonstrated that RAI treatment causes an elevation in ROS production after 6 h, which reaches its peak concentration at 48 h, and leads to a decrease in mRNA expression of sodium-iodide symporter (NIS) [114]. The augmented ROS production triggers an increase in both TrxR1 mRNA levels and enzyme activity, resulting in a reduction in OS. Various authors have concluded that the direct measurement of ROS concentrations could be a useful biomarker in evaluating clinical management [115,116]. This suggests that monitoring ROS production and TrxR1 levels could provide important information for optimizing RAI therapy and improving treatment outcomes for DTC patients. However, more research is necessary to comprehend the complete role of OS in RAI therapy and identify the most efficient approach for monitoring and managing OS in patients undergoing this therapy.

## 7. Antioxidants in Thyroid Cancer

Antioxidants function by safeguarding against ROS via the action of phytochemicals possessing antioxidant activity. Antioxidants are grouped into three categories based on their mechanism of action: (1) primary antioxidants that serve as free-radical scavengers or terminators, (2) secondary antioxidants that prevent chain initiation, and (3) tertiary antioxidants that participate in the restoration of impaired biomolecules [117,118]. Primary antioxidants include butylated hydroxyanisole (BHA), tertiary butylhydroquinone (TBHQ), propyl gallate (PG), and butylated hydroxytoluene (BHT), as well as natural antioxidants such as tocopherols, carotenoids, and flavonoids. These antioxidants have been observed to possess diverse biological activities, including antioxidant, anti-inflammatory, and anticancer properties. Flavonoids have been demonstrated to play a pivotal role in numerous events related to carcinogenesis, such as regulating cell growth and proliferation, inducing apoptosis, inhibiting angiogenesis, and modulating cellular signaling pathways. Numerous studies have shown the potential of flavonoids in the prevention and treatment of cancer. Moreover, several studies have reported that various flavonoids can induce apoptosis in TC cells by regulating the expression of pro- and anti-apoptotic proteins such as Bcl-2, Bax, caspases, and p53 [119,120,121]. For instance, quercetin has been demonstrated to activate caspase-3 and caspase-9 and down-regulate Bcl-2 expression, inducing apoptosis in PTC cells [121]. Additionally, flavonoids have been found to inhibit proliferation and induce apoptosis in various types of thyroid cancer cells, including PTC, FTC, and ATC. Luteolin, in particular, has been shown to upregulate Bax expression, down-regulate Bcl-2 expression, and activate caspase-3 and caspase-9 in ATC cells, resulting in apoptosis.

These findings suggest that flavonoids may hold therapeutic potential for treating TC [120]. In a study conducted by Leoni S. et al. on the FTC cell line PCCL3, flavonoids were found to inhibit the proliferation of TC cells by interfering with key enzymes involved in cell signaling and cell proliferation pathways, such as protein tyrosine kinase (PTK), protein kinase C (PKC), and DNA topoisomerases I and II [114]. Various flavonoids, including apigenin, biochanin A, chrysin, genistein, luteolin, and kaempferol, have been shown to effectively inhibit the proliferation of cell lines derived from FTC, PTC and ATC [119]. Additionally, some flavonoids such as resveratrol, quercetin, and genistein have been found to not only inhibit cancer initiation and progression, but also induce redifferentiation of TC cells [120]. Curcumin has been found to selectively promote cytotoxic effects in TC cells, but not in normal epithelial cells, and triggers autophagy by activating MAPK while inhibiting mTOR pathways. Abnormal activation of the AKT/mTOR axis is prevalent in PTC cases [121].

## 8. Future Perspectives

In the future, more specific and sensitive methods for measuring OS biomarkers may be developed. This could include the use of advanced mass spectrometry techniques or the development of more specific antibodies for targeting OS biomarkers. Additionally, further research is needed to better understand the relationship between OS and TC, as well as to identify specific pathways and targets for OS-related therapies. Personalized medicine approaches could also be developed to tailor OS-related treatments to individual patients based on their specific OS profile. Overall, continued research into OS biomarkers and their role in thyroid cancer could lead to the development of more effective and targeted therapies for this disease [122,123]. The standardization of methods used to assess OS biomarkers in TC is crucial for the accurate interpretation of results and comparison between studies. This can be achieved through the establishment of standardized protocols for sample collection, processing, and analysis. Additionally, the use of reference materials and controls can help to ensure the accuracy and reproducibility of the results. It is also important to consider the potential confounding effects of comorbidities and other factors that may affect OS biomarkers in TC patients. Further research is needed to identify the most reliable and specific OS biomarkers for TC, as well as to determine the optimal timing and frequency of OS biomarker measurements during treatment and follow-up [124].

One interesting future perspective in the therapy of TC is to explore new therapeutic methods based on the pathogenesis. Cheng et al. discovered that the expression of dipeptidyl aminopeptidase IV (DPP IV) is elevated in PTC and linked to the advancement of the disease, distant spreading, resistance to therapy, immune-evasive characteristics, and unfavorable clinical consequences [125]. Therefore, dipeptidyl aminopeptidase IV inhibitors (iDPP-IV) could be a new therapeutic target in the case of DTC.

Moreover, glycolysis as a metabolic pathway produces energy in cancer cells even in the absence of oxygen. As a consequence of this process, increased uptake of glucose and its metabolism, mediated by SGLT2 in a cancer cell, is higher than in normal cell and is connected with increased OS biomarkers [126]. Many studies have shown that SGLT2 is upregulated in different kinds of cancer [127,128]. Further investigation is needed to examine the possible use of SGLT2 inhibitors (SGLT2i) for treating patients with DTC, beginning with elucidating the underlying mechanisms that could support potential advantages and concluding with potential clinical ramifications.

Additional research is necessary to enhance comprehension and utilize the pathophysiological correlation between OS and the development of TC. Reactive oxygen species have a contradictory function in the occurrence and advancement of TC. Knowledge and comprehension of the molecular mechanisms that result in the development of TC provide significant possibilities for the creation of innovative therapeutic approaches.

## 9. Conclusions

Oxidative stress (OS) is known to play a significant role in the development and progression of TC. Understanding this relationship can pave the way for the development of new therapeutic strategies. Studies have indicated that higher levels of OS are associated with a worse prognosis in patients with TC. Therefore, more research is required to fully comprehend the underlying mechanisms and identify potential targets for antioxidant therapies that can reduce OS and enhance patient outcomes.

## Figures and Tables

**Figure 1 cancers-15-03182-f001:**
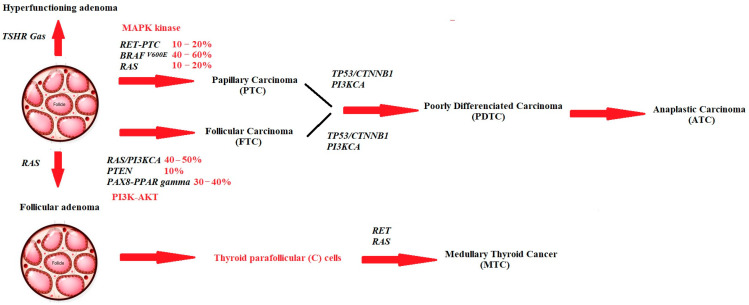
The thyroid carcinogenesis model encompasses the development of both benign and malignant tumors originating from the well-differentiated follicular cells of the thyroid. Autonomously hyperfunctioning thyroid adenomas are associated with activating mutations in the TSHR or Gαs genes. Moreover, differentiated thyroid cancer (DTC) can give rise to papillary thyroid carcinoma (PTC), follicular thyroid carcinoma (FTC), poorly differentiated thyroid carcinoma (PDTC), and anaplastic thyroid carcinoma (ATC) due to the acquisition of mutations in various oncogenes and tumor suppressor genes. Medullary thyroid cancer (MTC) arises from the parafollicular cells of the thyroid gland as a consequence of RET or RAS gene activation.

**Figure 2 cancers-15-03182-f002:**
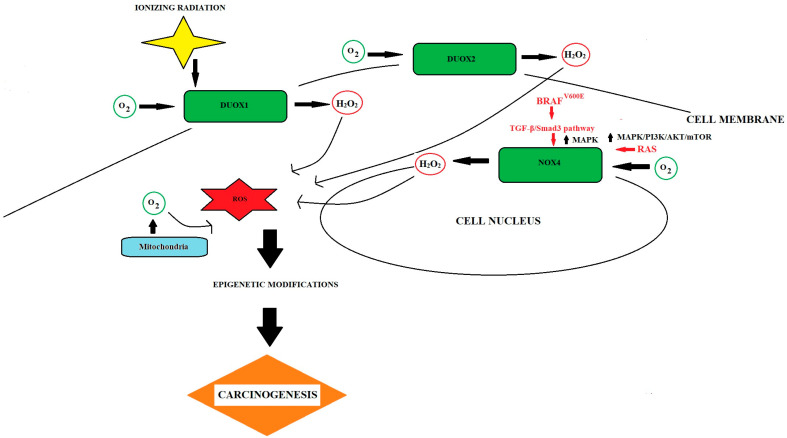
A diagram illustrating the impact of OS on TC development. Ionizing radiation, a recognized TC risk factor, induces the generation of H_2_O_2_ via DUOX1, resulting in DNA damage and potential genomic instability. NOX_4_, regulated by oncogenes BRAF**^V600E^** and RAS, is upregulated in TC. Mutations in the proto-oncogene BRAF increase NOX4 expression via the TGF-β/Smad3 pathway, resulting in the constitutive activation of MAPK. Oncogenic mutations in RAS also enhance NOX_4_-mediated H_2_O_2_ production and promote DNA damage.

**Figure 3 cancers-15-03182-f003:**
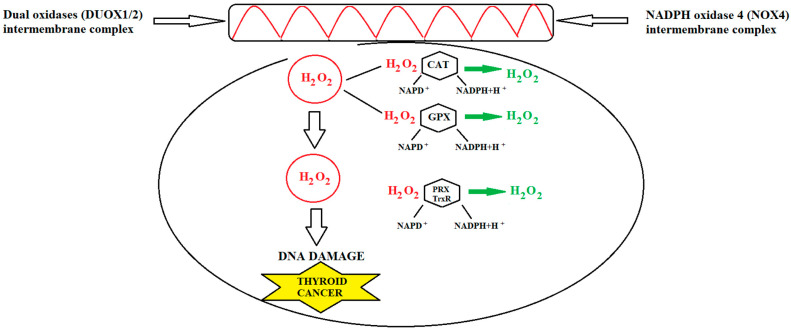
Antioxidant H_2_O_2_ elimination systems in the thyroid and the association of their impaired function with carcinogenesis development.

**Table 1 cancers-15-03182-t001:** ATA-risk stratification group in TC (2015) [58].

**ATA high-risk (>20%) category**	macroscopic invasion of tumor into the perithyroidal soft tissues; incomplete tumor resection; distant metastases; post-operative serum Tg suggestive of distant metastases; pathologic N1 with any metastatic lymph node ≥3 cm in largest dimension; follicular thyroid cancer with extensive vascular invasion (>4 foci of vascular invasion)
**ATA intermediate risk (5–20%) category**	microscopic invasion of tumor into the perithyroidal soft tissues; aggressive histology (e.g., tall-cell, hobnail-variant, columnar-cell carcinoma); PTC with vascular invasion; clinical N1 or >5 pathologic N1 with all N1 < 3 cm in largest dimension; multifocal papillary microcarcinoma with microscopic invasion of tumor into the perithyroidal soft tissues and BRAF^**V600E**^ mutation (if known).
**ATA low risk (<5%) category**	intrathyroidal PTC without vascular invasion, with or without small volume lymph node metastases (clinical N0 or ≤5 pathologic N1, all <0.2 cm in largest dimension); intrathyroidal encapsulated follicular variant of papillary thyroid cancer or intrathyroidal well-differentiated follicular cancer with capsular or minor vascular invasion (<4 vessels involved); intrathyroidal papillary microcarcinomas that are either BRAF WT or BRAF mutated (if known).

## Data Availability

Not applicable.

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
