# Peer review of "Could Oxidative Stress Play a Role in the Development and Clinical Management of Differentiated Thyroid Cancer?"

_cancers, 2023, doi:10.3390/cancers15123182_

Round 1

Reviewer 1 Report

I don't think it's a bad point of view, but I think it would be better to devise a little composition.

Abbreviations should be explained the first time they appear throughout.

Line 8 : 

Please insert the space. 

Simple Summary:Thyroid…… → Simple Summary: Thyroid……

Please remove period. 

……all cancer cases.. → ……all cancer cases.

Line 13: 

You must show the introduction DTC here. 

Line 37

Please check the space before 43,800.

In Reference

Please adjust the format according to the journal.

Ref. 1: You wrote Epub etc. Is that information need??

Ref. 3: It is difficult to understand this information. 

Ref. 3 to 6: Font is different from other references.

Ref. 8: I think this reference is inappropriate. The information described in the text cannot be known just by looking at this URL. 

Line 118, 124, 129, 228, 288, 293 and Figure 1

V600E should be changed to superscripted.

Or please stop using superscripts in the table.

Line 351

Please delete the space after DTC.

Line 384

Please delete the space after RAI. 

Line 437 to 453

What is OS marker. I don't know if there is the specific marker names and the description about “OS marker” is unclear.

Figure & Table

I think that Table1 should be devised to make it easier to see and understand for readers.

Please indicate in footnotes the references used to create the table.

Since this article is a review, it is better to use figures and tables to make it easier for readers to understand. I think it would be easier to understand if each section could be simply summarized.

If there are some illustrations about related genes, understanding would be deepened.

Author Response

Response to Reviewer 1 Comments

Point 1 : Line 8 :  Please insert the space. 

Simple Summary:Thyroid…… → Simple Summary: Thyroid……

Please remove period. 

……all cancer cases.. → ……all cancer cases

Response 1: Thank you for this suggestion. The changes have been made according to the reviewer's suggestion.

Point 2. Line 13: 

You must show the introduction DTC here. 

Response: Thank you for this valuable recommendation. The introduction section has been limited to the abstract section. Thus, the simple summary is containing

Point 3. Line 37

Please check the space before 43,800.

Response 3: The changes has been made according to the reviewer's suggestion. The space before 43,800 has been deleted.

Point 4: In Reference

Ref. 1: You wrote Epub etc. Is that information need??

Response : The revision has been made according to the reviewer's suggestion. The “Epub etc” has been deleted.

Ref. 3: It is difficult to understand this information. 

Response : The revision has been made according to the reviewer's suggestion. The reference 3 “Siegel RL, Miller KD, Fuchs HE, Jemal A. Cancer statistics, 2022. CA Cancer J Clin. 2022 Jan;72(1):7-33. doi: 10.3322/caac.21708.” has been added.

Ref. 3 to 6: Font is different from other references.

Response : The revision has been made according to the reviewer's suggestion. All references are written in Times New Roman font, size 10.

Ref. 8: I think this reference is inappropriate. The information described in the text cannot be known just by looking at this URL. 

Response : The revision has been made according to the reviewer's suggestion. Instead of the previously incorrect insert reference, a new one has been introduced “Megwalu UC, Moon PK. Thyroid Cancer Incidence and Mortality Trends in the United States: 2000-2018. Thyroid. 2022 May;32(5):560-570. doi: 10.1089/thy.2021.0662 „

Point 5. Line 118, 124, 129, 228, 288, 293 and Figure 1

V600E should be changed to superscripted. Or please stop using superscripts in the table.

Response : The revision has been made according to the reviewer's suggestion. In line 118, 124, 129, 228, 288, 293 and Figure 1 V600E has been changed to a superscript format.

Point 6. Line 351 Please delete the space after DTC.

Response : The revision has been made according to the reviewer's suggestion. In line 351 the space after DTC has been deleted.

Point 7. Line 384 Please delete the space after RAI. 

Response : The revision has been made according to the reviewer's suggestion. In line 384 the space after RAI has been deleted.

Point 8. Line 437 to 453 What is OS marker. I don't know if there is the specific marker names and the description about “OS marker” is unclear.

Response : Since oxidative stress is impossible to measure directly, many oxidative stress-related proteins have been implicated as `oxidative stress marker` to indirectly measure oxidative stress status. Thus, in lines 437 to 453, the term "OS marker" has been changed to "OS biomarker". The term biomarker has been defined by The National Institutes of Health as "a characteristic that is objectively measured and evaluated as an indicator of normal biological processes, pathogenic processes, or pharmacological responses to a therapeutic intervention`. The role of oxidative stress in the pathophysiology of thyroid cancer is well established, where `oxidative stress biomarker` is frequently used among literature data. Reactive oxygen species (ROS) are derived from various sources, including mitochondria, xanthine oxidase, uncoupled nitric oxide synthases, and NADPH oxidase. In addition to causing generalized oxidation leading to cell dysfunction, necrosis, or apoptosis, ROS also induce specific post-translational modifications that alter the function of important cellular proteins and signaling pathways in the thyroid. Biomarkers of oxidative stress can be classified as molecules that are modified by interactions with ROS in the microenvironment, as well as molecules of the antioxidant system that change in response to increased redox stress. Examples of molecules that can be modified by excessive ROS in vivo include DNA, lipids (including phospholipids), proteins, and carbohydrates. Nevertheless, these oxidatively modified biological components concentrations may be use as indirect oxidative stress markers.

Figure & Table

I think that Table1 should be devised to make it easier to see and understand for readers.

Please indicate in footnotes the references used to create the table.

Since this article is a review, it is better to use figures and tables to make it easier for readers to understand. I think it would be easier to understand if each section could be simply summarized.

If there are some illustrations about related genes, understanding would be deepened.

Response : Thank you for these valuables recommendations. The figure 1 has been changed following Reviewer`s suggestion and necessary summarizing of subsequent Review chapters have been added. Furthermore, the Figure 2 describing the role of H2O2 in thyroid hormone synthesis and carcinogenesis development has been added.

Reviewer 2 Report

The manuscript of Kosciuszko et al. is aimed at summarizing the current knowledge concerning the role of oxidative stress (OS) in thyroid cancers. In specific, Authors aim to review the literature providing information about OS in the development of thyroid cancers and the possible therapeutic options to employ OS measurement in the assessment of thyroid cancer risks and to pharmaceutically target OS. The review is well constructed, and provides a succinct overview of this complex field. It contains a general “Introduction” followed by chapters describing the diagnosis, molecular pathogenesis and treatment options of thyroid cancers. In addition, the last two chapters focus specifically on the role of OS in radiotherapy and anti-oxidants as therapeutic options. The manuscript is supported by 1 Figure and 1 Table and cites 128 references.  

The manuscript fits the scope of the “Cancers” and is of interest for the readers of the journal. The manuscript is well written and provides a balanced review of the literature.

This reviewer notes the following issues that need to be addressed before the manuscript could be considered for acceptation.

1.     Current expert opinion concerning oxidant-related signaling recommends mentioning the exact oxidant type (eg. hydrogen peroxide, superoxide) whenever it is possible or if mentioned in a generalized fashion refer to them as “oxidants” instead of “reactive oxygen species (ROS)” (see Sies et al.  https://doi.org/10.1038/s41580-022-00456-z). Please revise the text accordingly; it is essential to promote the most current and precise scientific view in oxidant-related research. 

2.     Hydrogen peroxide produced by the DUOX2 enzyme is critical for thyroid hormone biosynthesis. An additional Figure or Table with a summary of aberrant function of oxidant sources in relation to carcinogenesis would be very helpful.

3.     Thyrocytes are singularly resistant to oxidative stress. In this context it would be necessary to provide more details about the antioxidant systems and their relation to the carcinogenic factors mentioned in Figure 1.

4.     Figure 1: please provide a more detailed description in the Figure Legend and include the abbreviation in this text.

5.     Page 7: Ref. 86 is a review, please revise the sentence accordingly and mention it as (reviewed in 86).

6.     Table 1: please check the font types: the “ATA high” category is in bold character while the other parts are not.

Author Response

Response to Reviewer 2 Comments

Point 1.     Current expert opinion concerning oxidant-related signaling recommends mentioning the exact oxidant type (eg. hydrogen peroxide, superoxide) whenever it is possible or if mentioned in a generalized fashion refer to them as “oxidants” instead of “reactive oxygen species (ROS)” (see Sies et al.  https://doi.org/10.1038/s41580-022-00456-z). Please revise the text accordingly; it is essential to promote the most current and precise scientific view in oxidant-related research. 

Response: Referring to the recommended literature data, the mentioned in the manuscript theme `ROS` is provided correctly, due to the oxidative modified biological component measurement among cited studies. The manuscript was revised accordingly Reviewer recommendation and no oxidants have been found.

Point 2.     Hydrogen peroxide produced by the DUOX2 enzyme is critical for thyroid hormone biosynthesis. An additional Figure or Table with a summary of aberrant function of oxidant sources in relation to carcinogenesis would be very helpful.

Response: Accordingly to the Reviewer suggestion, the Figure 2 illustrating the impact of hydrogen peroxide, DUOX2, and NOX4 enzyme on thyroid cell dysfunction with the development of the tumorigenic process has been added.

Point 3 .     Thyrocytes are singularly resistant to oxidative stress. In this context it would be necessary to provide more details about the antioxidant systems and their relation to the carcinogenic factors mentioned in Figure 1.

Response: Thank you for this recommendation. The section concerning the antioxidant-related factors has been implicated.

Point 4.     Figure 1: please provide a more detailed description in the Figure Legend and include the abbreviation in this text.

Response: Figure 1 has been modified to clearify data and subsequent legend has been added accordinlgy to the valuable Reviewer suggestion.

Point 5.     Page 7: Ref. 86 is a review, please revise the sentence accordingly and mention it   

Response: The sentence has been revised accordingly to Reviewer reccomendation.

Point 6.     Table 1: please check the font types: the “ATA high” category is in bold character while the other parts are not.

Response: Thank you for your suggestions, the suggested changes have been made.

Round 2

Reviewer 1 Report

Please check following points; 

Line 45-6: For easy reading, it is better for you to insert space. 

Line 79-80: For easy reading, it is better for you to insert space. 

Line 160: There is a strange punctuation.

Line 169: Aren't footnotes supposed to end with a period?

Line 211-2: For easy reading, it is better for you to insert space. 

Table 1:  The table is difficult to see, so it is better to align the characters to the right, not evenly space them.

Line266: H2O2 should be H2O2 (subscript).

Figure2: H2O2 should be H2O2 (subscript).

Line270: H2O2 should be H2O2 (subscript).

Line 310-1: For easy reading, it is better for you to insert space. 

Line 352-3: For easy reading, it is better for you to insert space. 

Line 380-1: For easy reading, it is better for you to insert space. 

Line537: Why is there a space here?

English is not my native language. I don't think there are any particular grammatical problems in this article, but it was a little hard to read.

Author Response

Response to Reviewer 1 Comments

Line 45-6: For easy reading, it is better for you to insert space. 

Response: Thank you for this suggestion. The changes have been made according to the reviewer's suggestion

Line 79-80: For easy reading, it is better for you to insert space. 

Response: Thank you for this suggestion. The changes have been made according to the reviewer's suggestion

Line 160: There is a strange punctuation.

Response: Thank you for this suggestion. The changes have been made according to the reviewer's suggestion

Line 169: Aren't footnotes supposed to end with a period?

Response: Thank you for this suggestion. The changes have been made according to the reviewer's suggestion

Line 211-2: For easy reading, it is better for you to insert space. 

Response: Thank you for this suggestion. The changes have been made according to the reviewer's suggestion

Table 1:  The table is difficult to see, so it is better to align the characters to the right, not evenly space them.

Response: Thank you for this suggestion. The changes have been made according to the reviewer's suggestion

Line266: H2O2 should be H2O2 (subscript).

Response: Thank you for this suggestion. The changes have been made according to the reviewer's suggestion

Figure2: H2O2 should be H2O2 (subscript).

Response: Thank you for this suggestion. The changes have been made according to the reviewer's suggestion

Line270: H2O2 should be H2O2 (subscript).

Response: Thank you for this suggestion. The changes have been made according to the reviewer's suggestion

Line 310-1: For easy reading, it is better for you to insert space. 

Response: Thank you for this suggestion. The changes have been made according to the reviewer's suggestion

Line 352-3: For easy reading, it is better for you to insert space. 

Response: Thank you for this suggestion. The changes have been made according to the reviewer's suggestion

Line 380-1: For easy reading, it is better for you to insert space. 

Response: Thank you for this suggestion. The changes have been made according to the reviewer's suggestion

Line537: Why is there a space here?

Thank you for your suggestion. The space in line 537 has been deleted, as per the reviewer's recommendation.

Reviewer 2 Report

The manuscript of Kosciuszko et al. is has been substantially revised, however, there are still some issues that need to be addressed to be considered for publication.

1.     Hydrogen peroxide produced by the DUOX2 enzyme is critical for thyroid hormone biosynthesis. An additional Figure or Table with a summary of aberrant function of oxidant sources in relation to carcinogenesis would be very helpful.

Authors simply modified Figure 2 without mentioning the specific association of NOX4 with BRAF mutated thyroid malignancies and to indicate the different location of H2O2 production by DUOX 2 and NOX4. Please prepare a more detailed figure. 

2.     H2O2 should be corrected for H2O2.  In general, the manuscript needs to be revised for mistyping errors but this is the duty of the Author and is not for the reviewer to cite every example.

Author Response

Response to Reviewer 2 Comments

1.Hydrogen peroxide produced by the DUOX2 enzyme is critical for thyroid hormone biosynthesis. An additional Figure or Table with a summary of aberrant function of oxidant sources in relation to carcinogenesis would be very helpful.

Accordingly to the Reviewer suggestion, the modified Figure 3 illustrating the impact of H2O2, DUOX2, and NOX4 enzyme on thyroid tumorigenic process has been added.

Authors simply modified Figure 2 without mentioning the specific association of NOX4 with BRAF mutated thyroid malignancies and to indicate the different location of H2O2 production by DUOX 2 and NOX4. Please prepare a more detailed figure. 

Accordingly to the Reviewer suggestion the new figure illustrating association of NOX4 with BRAF and RAS mutated thyroid cancers has been added (Figure 2).

  1. H2O2 should be corrected for H2O2. In general, the manuscript needs to be revised for mistyping errors but this is the duty of the Author and is not for the reviewer to cite every example.

Response: Thank you for your suggestions, the suggested changes have been made.
